# A Paleopathological Find on a La Tène Horse Skeleton Discovered in Rescue Archaeological Diggings in the Area of the Olympic Pool, Alba Iulia (CX 143 Pit)

**DOI:** 10.3390/ani14121775

**Published:** 2024-06-13

**Authors:** Alexandru Gudea, Florin Gheorghe Stan, Alexandra Irimie, Sorin Mârza, Cristian Olimpiu Martonos, Adrian Gal

**Affiliations:** 1Faculty of Veterinary Medicine Cluj-Napoca, University of Agricultural Sciences and Veterinary Medicine Cluj-Napoca, 400372 Cluj-Napoca, Romania; florin.stan@usamvcluj.ro (F.G.S.); alexandra.irimie@usamvcluj.ro (A.I.); sorin.marza@usamvcluj.ro (S.M.);; 2School of Veterinary Medicine StKitts and Nevis, Ross University, Basseterre P.O. Box 334, Saint Kitts and Nevis; cmartonos@rossvet.edu.kn

**Keywords:** paleopathology, La Tène, Late Iron Age, horse, scapula, periosteal lesion, Transylvania, Romania

## Abstract

**Simple Summary:**

This paper presents, to the best of our knowledge, the first documented instance of a specific pathological lesion found in a Late Iron Age (La Tène period) horse. This discovery was made in the CX 143 pit in Alba Iulia, Romania, during archaeological rescue excavations conducted in 2022. In addition to the standard morphological features of the horse, comprehensive investigations revealed an unusual lesion on the left scapular neck. This lesion was evaluated both macroscopically and radiologically and has been interpreted as periarticular and supraarticular hyperostosis. A series of hypotheses were proposed regarding the cause of this specific type of injury, including the possibility of it being traumatic in nature, and potentially influenced by particular conformational aspects of the identified horse.

**Abstract:**

The reconstruction of past life based on archaeozoology is a challenging domain that offers a range of valuable details concerning former human and animal populations. Additionally, the ancient era is a source of information for human and veterinary medicine, as well as for other biological sciences. This report highlights a pathological lesion identified during the investigation of a horse skeleton from a pit dated La Tène in Alba County (Romania). The left scapula with lesions was collected from the skeleton of a buried 7–8-year-old male horse. The aforementioned bone underwent gross, radiological, and computerized tomography evaluation. Macroscopically, a collar of supraarticular cancellous hyperostosis was detected, most likely as a consequence of an invasive chronic phlegmonous periarthritis and/or bursitis of the infraspinate muscle following probably a penetrating cutaneous wound in the shoulder region. A suppurative periarthritis\bursitis of the infraspinate muscle situated nearby caused, apparently, supraglenoidian periosteitis responsible for osteophytes and exostoses formation in the neck region of the scapula.

## 1. Introduction

Along with regular archeozoological studies, paleopathology may also reveal different diseases of the ancient period and may serve as a source of knowledge for medicine, veterinary medicine, and other biological sciences, including the history of disease [1,2,3]. Paleopathology is an important and growing field, though challenging as well, as evidence for disease is limited to what part of the organism is preserved. Data extracted from the study of the remaining bones are usually scarce, as not so many diseases are reflected in bone lesions. [2].

### 1.1. Brief Description of the Historical and Archaeological Context

The La Tène period and culture is a period belonging to the European Iron Age Culture. The chronology of this period is from about 450 BC to the Roman Conquest (1st century BC) and corresponds to the nowadays territories of France, Germany, Belgium, Switzerland, Austria, Hungary, Czech Republic, Slovenia, Hungary, and some parts of Romania, more precisely the western part of the country (Transilvania) and Western Ukraine [4]. The subject of our investigation was discovered in Alba county, Romania in CX 143 burial pit in the area of the Olympic Pool, Alba Iulia that contains a complete horse skeleton that was earlier analyzed and published [5]. The morphological characteristics revealed by the previous analysis indicate a 7–8-year-old male, with a height of 1200–1300 mm, which was calculated based on the composite formula of Bartosiewicz [6], similar to other horse skeletons discovered in the same area [7,8,9]. These other findings did not show significant pathological or subpathological lesions, making the present report a singular one, given the historical context for this territory.

### 1.2. Anatomical Details on the Distal Scapular and Proximal Humeral Region in the Horse and the Osteo-Musculo-Articular Lesions in Horse

The scapula (*scapula*) in the horse is the upper part of the forelimb skeleton that ensures the connection of the trunk with the appendage. The lateral surface of the bone (*facies lateralis*) and the articular angle bears the scapular spine (*spina scapulae*) and the spine tuberosity (*tuber spinnae scapulae*) delimiting the supraspinous (*fossa supraspinata*) and subspinous fossae (*fossa infraspinata*). The neck of the scapula (*collum scapulae*) continues distally with the glenoid cavity (*cavitas glenoidalis*) and the supraglenoidian tubercle (*tuberculum supraglenoidale*). The humerus (*humerus*) shows on its proximal end the evident humeral head (*caput humeri*) with a neck (*collum humeri*) and the existence of the greater (*tuberculum majus*), lesser (*tuberculum minus*) and intermediary tubercles (*tuberculum intermedium*) that form the intertubercular groove (*sulcus intertubercularis*) [10].

The muscular structures associated with the lateral scapular surfaces of the scapula are the deltoideus muscle (*m. deltoideus*) which is the most superficial one; the supraspinate muscle (*m. supraspinatus*) and the supraspinate synovial bursa (*b. subtendinea m. supraspinati*) present at the level of the passage over the scapular tuberosity; the infraspinate muscle (*m.infraspinatus*) and the infraspinate bursa (*b. subtendinea m. infraspinati*) is synovial pouch interposed between the wide tendinous part and distal scapular surface and proximal humeral structure; and the teres minor muscle (*m.teres major*). Other related structures include the coracobrachialis muscle (*m. coracobrachialis*) and its bursa (*b. subtendinea m. coracobrachialis*) and the biceps brachii muscle (*m.biceps brachii*) and its bicipital bursa (*intertubercular bursa*) (*vag. synovialis intertubercularis*) [11,12,13].

Among the pathologies encountered at the distal level of the scapula and the proximal humeral part, nowadays the veterinary pathology literature mentions the following commonly encountered conditions [14]:Osteoarthritis and osteochondritis of the shoulder joint [15];Fractures and other mechanical injuries, muscular atrophy, most frequently of secondary origin -the sweeny shoulder [16];The infraspinate tendon and its bursa [17,18];Intertubercular (bicipital) bursitis [19].

### 1.3. Paleo-Pathology of the Ancient Horses Revealed by Osteologically Identifiable Lesions

As horses seemed to play a very important role in the life of human ancestors through time [20], they provide a special opportunity for the study of pathological lesions seen in the skeletons or bone pieces. One advantage is the fact that horses were not regularly used as a source of meat [21,22], but most often as draught animals, pulling chariots and wagons, or as battle animals, and their remains stayed quite undamaged by the usual anthropic factors. However, as typically only bones are available for study, a record of the pathological changes is limited by the amount of visible damage expressed onto the identified skeletal pieces. This is the reason that paleopathological data are quite scarce in the specialized literature. If we focus on a more specific and narrower timespan (such as the one implied here) the paleopathology elements become a very rarely documented fact with quite a few elements relevant to the health status of a given animal population.

Previous investigations have suggested that some archaeological horse remains bear osteological changes associated with horseback riding [20,23,24,25]. There are sets of morpho-pathological cues that are associated with supplementary stress on the osteo-muscular system. Some of them include osteopathic lesions (spondyloarthropathy) in the thoracic and lumbar spine that may lead to intervertebral fusion with different degrees of clinical manifestations, mechanical injuries (associated with intense labor) that lead to ossifications in the metacarpal and metatarsal bones [26] or in the phalangeal segments (ringbones—chronic ossifying periostitis), including here the ossification of complementary fibrocartilages. There is limited discussion regarding paleopathological findings related to other cranial pathologies, such as the effects of additional features of horseback riding. For instance, the use of a bit can result in diastemal lesions and distinctive enamel wear patterns, particularly on the anterior surface of the first premolar. [24].

Other pathological conditions that have osteological expression might include tarsal and carpal degenerative joint diseases (such as spavin) and several conditions (including infectious ones and of traumatic origin) that lead to a periosteal response on various bony elements [26,27]. Animal paleopathology reports rarely mention bone lesions associated with nutritional and metabolic disorders (MDB—metabolic bone disease) [28,29].

Our report details a pathological condition identified in the reconstructed scapula of a horse skeleton unearthed from a pit dating back to the La Tène period in Alba County, Romania, during archaeological excavations conducted in 2023. There is no prior reference in the literature in terms of paleopathology for the region of the shoulder joint in the horse, so this makes this study a singular one, pointing to a new find in this respect.

## 2. Materials and Methods

The excavated material was presented to the Anatomy Lab of the Faculty of Veterinary Medicine Cluj-Napoca for an in-depth examination. The sample was analyzed, and various usual archeozoological assessments were made (ossification stages, osteometrical, numeric) that led to the conclusions mentioned in the introductory part. The examined piece, the right scapula, was reconstructed (as the material was very frail and highly degraded due to exposure to soil acidity) so the articular angle of the bone was identifiable and explorable. One must mention the fact that the left scapula was not displaying any pathological signs.

A macroscopic examination was conducted initially by an archeozoologist and then by a specialized veterinary pathologist who evaluated the extent and quality of the observed lesions. Next, the scapular fragment was examined by using X-ray radiography at the Laboratory of Radiology and Medical Imaging, Faculty of Veterinary Medicine of Cluj-Napoca. The X-ray images were made using a fixed radiographic device TEMCO Grx-01 (K&S Röntgenwerk Bochum GmbH&Co KG, Bochum, Germany). The exposures were made dorsoventrally, on both (lateral and medial) perspectives. The parameters used to obtain the images were 50–56 kV and 13–20 mAs. The images were acquired using a DR Flat Panel detector Reyance Xmaru 1717SGC/SCC (Reyance Inc., Hwaseong-si, Gyeonggi-do, Republic of Korea) and Xmaru VetView V1 (Reyance Inc., Hwaseong-si, Gyeonggi-do, Republic of Korea) acquisition software.

Helical CT scanning of the scapula was obtained using a Siemens CT Somatom Scope machine with 16 channels (Faculty of Veterinary Medicine, Cluj-Napoca, Romania). The scan was made with bones on the medial and lateral sides with parameters kVP 130 and mAs: 54. Bone images were obtained in the axial plane using a 512 × 512 matrix, narrow windows (WW: 120, WL: 40), 3 mm slice thickness, with a pitch of 1.5. Multiplanar image reconstruction of the bones was obtained using soft tissue and bone window reconstruction at a slice thickness of 1 mm. A specialized clinician of the Imagistics Department of our Faculty helped in interpreting the resulting data.

## 3. Results

### 3.1. Description of the Macroscopic Investigation

The gross features (Figure 1) detected in the distal portion of the scapula (neck of the scapula) include a major hypertrophy of the region due to the formation of an exuberant osseous tissue.

The newly formed bone tissue displays an irregular surface just dorsally (2–3 mm) to the glenoid cavity (Figure 1), all around the neck of the scapula. Consequently, the infraglenoid tubercle and the coracoid process are barely visible, being embedded in the osteophytes/exostoses situated in the neck of the scapula. The osteophytes and exostoses are most likely of cancellous bone, a fact suggested by the X-ray images (Figure 2).

### 3.2. Description of the Imagistic Investigation

The radiological investigation showed (Figure 2), as expected, just densification of the bony structures around the scapular neck, with no possibility of a clear link to a pathological situation. There is a lack of differentiation among the neighboring structures, such that the supraglenoid tuberosity and its usually much denser structure are hard to tell apart from the surrounding tissue. The borderline between the denser, newly deposited bone is visible also on the radiographic images, especially on the lateral perspective, marking the lateral trajectory of the suprascapular vessels and nerves.

The analysis of the CT images (Figure 3 and Figure 4) indicates a crack that affects the medial blade (which is the broken reconstructed piece), that starts beneath the glenoid cavity, extending along the articular angle, and spanning the entire width of the back’s neck towards the thoracic angle. On the lateral surface, the lack of the dorsal spine can be observed, with the base of the dorsal spine preserved on the cranial edge.

The compact osseous tissue is very well distinguished, both on the lateral side with an average Hounsfield unit of 961, and on the medial side, with an average Hounsfield unit of 1012. The spongy tissue is almost completely absent, with a small presence at the neck level, with an average Hounsfield unit of 98. The spans are visible towards the dorsal edge. These features indicate a limited topical reactive and hypertrophic bony process confined to the superficial level of the compact bone, which did not involve a reactive and proliferative process towards the cancellous core part of the bone.

## 4. Discussion

The structure under examination encompasses the articular angle of a scapula, featuring a visible neck, glenoid cavity, and the supraglenoid tubercle. Macroscopic examination revealed a collar of newly deposited bony material in the form of spicules (periarticular hyperostosis). The bone structure, especially at the level of the articular angle, is comprised of a production of amorphous bone tissue with an average of 199 Hounsfield units. This structure does not affect the lamellae of the bone tissue and can be interpreted as hyperplastic periostitis.

Regarding the fundamental mechanism of periosteal lesion induction, only the outer layer of the bone, known as the cortical bone, is affected. This differs from osteomyelitis, where the bone marrow cavity and associated cancellous bone are also involved.

In the presented case, only the outer part of the compact bone is affected, as observed on X-ray and CT scan images.

During prolonged chronic periosteal injury (chronic periostitis), the osteogenic layer of the periosteum responds to the injury by eventually forming cancellous bone on the surface of the compact cortical bone as benign bony overgrowths, known as osteophytes and exostoses (as seen in our case). Since no muscle has significant insertion points in the hypertrophied area (with osteophytes/exostoses), enthesophyte formation typical for tendon or ligament damage is absent. Given that the osteophytes and exostoses are located surrounding the neck of the scapula (just dorsal to the glenoid cavity), phlegmonous suppurative inflammation is one of the few conditions with an invasive nature (attributable to the degrading enzymes of micro and macrophages) that could explain this. This pathology helps to elucidate the periostitis extending, possibly, on both sides of the scapula’s neck.

Regarding the etiology of the suppurative periarthritis that ultimately leads to supraglenoid periostitis (essentially the primary cause of osteophyte/exostosis formation), the suppurative bacteria could have reached the affected area either hematogenously (from sites other than the leg region) or through a regional onset, presenting another possible scenario.

We lean towards the latter option because the bursa of the infraspinatus muscle is situated directly above the affected region. In terms of the occurrence of suppurative bursitis of the infraspinatus muscle, the most probable method of bacterial seeding into the bursa could be through a puncture or cut wound in the scapulohumeral joint region. In the historical period assessed, the above-mentioned joint region was most likely free of overlapping harnesses [30,31] that could cause infection by skin erosion.

In the current literature [32], this type of bursitis is frequently associated with individuals with narrow chests (so with closely placed forelimbs) used for fast trotting (as it was, most probably the individual from La Tene period, being 1200–1300 mm high, with slender extremities [5]) or may be caused by severe adductive movements (causing mechanical trauma). The bursitis might have been of a non-septic type (and initiated also by other traumatic factors) but most probably rapidly turned into a septic one. Since in the La Tène period, horses played a crucial role as draught animals, pulling chariots and wagons, or as battle animals, the individuals were quite exposed to a range of injuries, including puncture or cut wounds. The initial traumatic event (logically deduced from the historical time and context) may have also been a laceration or a puncture wound (potentially from an arrowhead or spearhead) that did not result in the animal’s death. Instead, it likely evolved into a locally confined septic process, facilitated by the hard bony surface of the scapula.

Regardless of the initial cause of the lesion (conformational, traumatic, or septic), symptomatology, like the one in bicipital bursitis, is usually mild, with animals standing with the leg partially abducted. The stride, in most cases, is decreased in its cranial phase [17,32] and does not exclude the animal from its usual functions.

## 5. Conclusions

Archaeozoology combines equally delightful domains, specifically archaeology and zoology, that analyze animal remnants from archaeological sites. The records of the bone lesions performed by the faunal analysis on animal fragments are usually scarce.

Our latest findings indicate the presence of a pathological condition in the reassembled right scapula of a male horse skeleton (7–8 years old) discovered in a La Tène-era pit in Alba County, Romania. This condition appears to be identified for the first time based on our current knowledge. The skeleton was excavated during archaeological diggings in 2023.

Gross examination revealed a collar of supraarticular hyperostosis made of cancellous bone, caused, most probably, by an invasive chronic inflammatory lesion situated in the proximity of the scapulohumeral region. A range of chronic disorders could be responsible for osteophytes and exostoses formation in the scapular neck, for instance, the collar-like distribution of the benign bony overgrowth could be the consequence of phlegmonous periarthritis and associated supraglenoidian periostitis. However, a more probable cause for supraglenoidian periostitis would be the phlegmonous bursitis of the infraspinate muscle, which coincides with the affected zone. Additionally, similar reports from zooarchaeologists are required to furnish more detailed information concerning the pathologies associated with animal populations in past life. 

## Figures and Tables

**Figure 1 animals-14-01775-f001:**
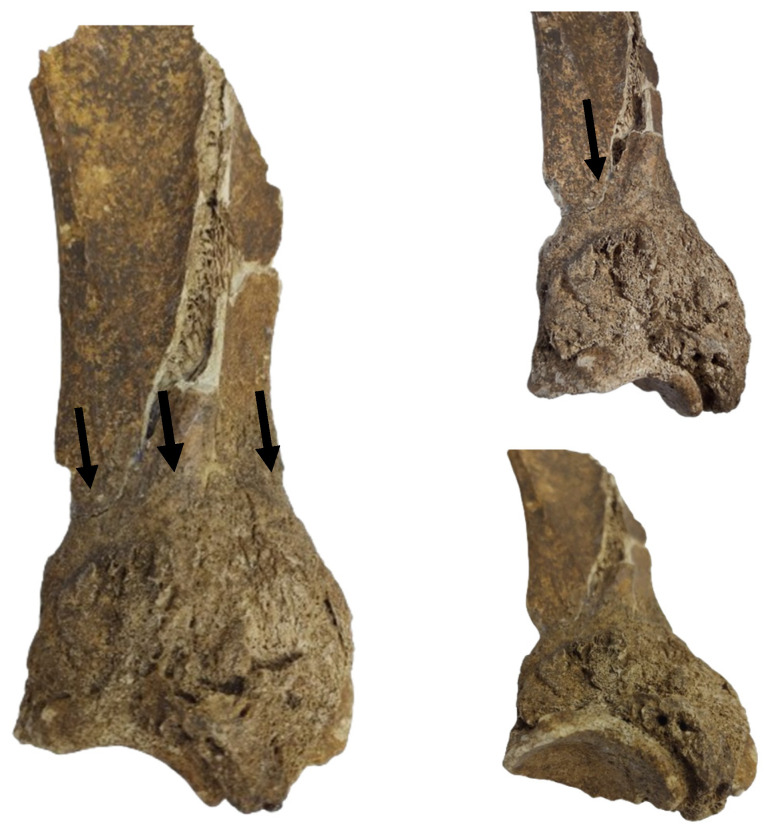
The scapular fragment. The aspect of the lesion onto the lateral surface of the scapular neck. Note the development of osseous tissue and the marking of the suprascapular artery and nerve trajectory (arrow).

**Figure 2 animals-14-01775-f002:**
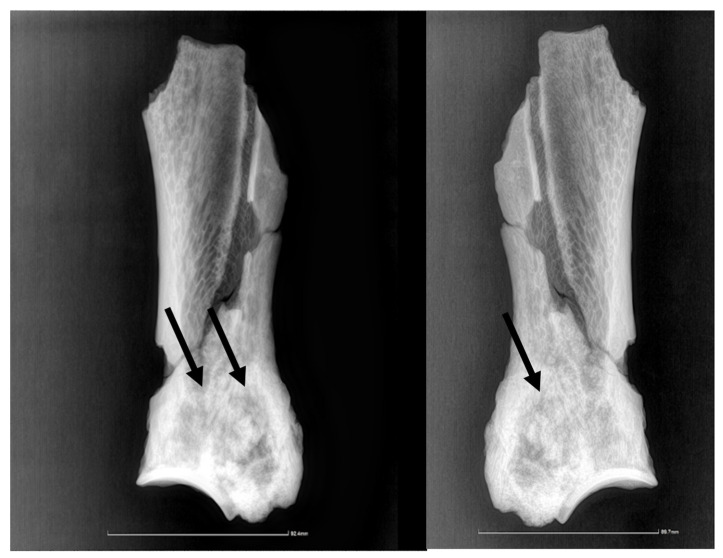
The radiographic images in lateral and medial perspectives. The arrow points to the delimitation line between the newly deposited bone and the compact bone of the lateral scapular surface.

**Figure 3 animals-14-01775-f003:**
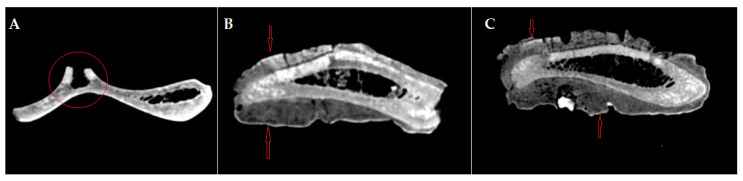
CT images of the scapula: (**A**) axial section of the scapula, preserving the base of the dorsal spine on the cranial edge—red circle; (**B**,**C**) axial section of the scapula in the neck region of the scapula, intercepting the proliferated bone on both sides, lateral and medial—red arrows.

**Figure 4 animals-14-01775-f004:**
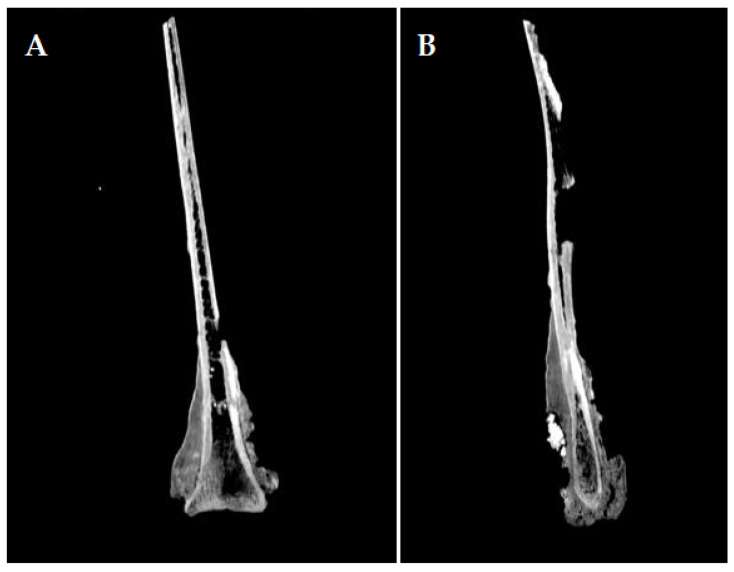
CT images of the scapula: (**A**) sagittal section of the scapula displaying the extending tendency of osteophytes and exostoses on the scapula; (**B**) sagittal section of the scapula showing the hyperostotic bone tissue.

## Data Availability

The original contributions presented in the study are included in the article material, further inquiries can be directed to the corresponding author/s.

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
