# Peer review of "A Paleopathological Find on a La Tène Horse Skeleton Discovered in Rescue Archaeological Diggings in the Area of the Olympic Pool, Alba Iulia (CX 143 Pit)"

_animals, 2024, doi:10.3390/ani14121775_

Round 1
Reviewer 1 Report
Comments and Suggestions for Authors
From my perspective and although I am not a specialist in paleopathology analysis, I think that the article is original, interesting and that it addresses aspects little covered in the literature. Therefore, for me it is a publishable article with minor comments. I only have one issue to highlight and that is that the discussion seems too short to me. I think the authors could expand the discussion and make more reference or at least discuss a little more about the causes that caused the pathology.
Author Response
Dear Reviewer,
Thank you very much for your notes on our paper. As mentioned in your review, some of the elements of the Results section were moved to the Discussions part. Some other additions in terms of differential diagnosis were added to this part. A more structured discussion on the possible causes of the lesion has been created. Differential diagnosis data is inserted in the last part of the paper.
Reviewer 2 Report
Comments and Suggestions for Authors
me appreciate reading your article. me found a valuable contribution to teh study of ancient animal remains.
me provide additional comments in teh attached file

Teh text is very clear, but I noticed a kind of 'scientific way' of writing: lots of brackets, enumeration of facts, sentences not always clearly connected, and use of words like 'thus' which I is not sure is appropriate for a written text. I is sure that some editing would improve teh quality of teh text.
Author Response
The text was checked once again by a professional. Several (sometimes significant) changes, both grammatically and semantically were operated. Some parts of the initial text were also removed and adapted to the reviewer’s demands. Thank you for your observations!
Reviewer 3 Report
Comments and Suggestions for Authors
Paleopathology is a growing and important field and very illuminative for our knowledge of both animal husbandry and animal pathologies in the past. This paper adds to our knowledge and presents a unique pathological occurrence for an equine.
I do think a partial rewrite would be good - moving some of the material located in the results to the discussion, and in particular providing a bit more clarity in two areas.
1. Please go into more detail as to your differential diagnosis. You present two possible pathological conditions, or rather causes for the observed osteoperiostitis) and then seem to rank one as more probable (bursitis of the infraspinate muscle as opposed to periarthritis)? Why? What observed features lead to this conclusion? What observations of different areas of scapula support this: joint (deformed or not), bone (enthesophyte?), periosteum (lesion size?), etc.? That is, on what basis, direct or indirect are these causes discriminated? Also here try to use best overall label for pathological condition (osteoperiostitis ?).
2. Also please go into more detail as to your preferred explanation for the external cause of this pathological condition. Other papers attribute bulk of lesions to overuse of horses by humans (riding). You instead find trauma a more likely cause it seems (warfare ?). Again, on what basis? What observed details suggest this?
I do think, however, the paper is important and worth publishing. I just think more evidence needs to be given linking osteological observations to pathological internal and external causes.

I noted a few changes that should be made in attached document. Overall English fairly clear, though I sometimes got the impression standard English veterinary or anatomical terms were not properly used (see comments in paper).
Author Response
Paleopathology is a growing and important field and very illuminative for our knowledge of both animal husbandry and animal pathologies in the past. This paper adds to our knowledge and presents a unique pathological occurrence for an equine.
- I do think a partial rewrite would be good - moving some of the material located in the results to the discussion, and in particular providing a bit more clarity in two areas.
Done as suggested in pdf file
- Please go into more detail as to your differential diagnosis. You present two possible pathological conditions, or rather causes for the observed osteoperiostitis) and then seem to rank one as more probable (bursitis of the infraspinate muscle as opposed to periarthritis)? Why? What observed features lead to this conclusion? What observations of different areas of scapula support this: joint (deformed or not), bone (enthesophyte?), periosteum (lesion size?), etc.? That is, on what basis, direct or indirect are these causes discriminated? Also here try to use best overall label for pathological condition (osteoperiostitis ?).
The text went through a re-arrangement, based on the observations you made in the attached pdf file, especially on the discussion and conclusion part. Elements for differential diagnosis were added. Concerning the possible suppurative bursitis or periartritis for osteophytes and exostoses formation, we presume both of them could be responsible for supraglenoidian periosteitis. We cannot exclude one from the other. For example, a supurative bursitis may lead to periatritis by necrotizing the bursal sacs by the purulent material. However, vice versa is also possible. As a consequence, we suppose the suppurative bursitis and/or periartritis should be used together in the occurred situation. More details regarding the full “pathological portrait” we presume is described below and in the body text of the paper.
- Also please go into more detail as to your preferred explanation for the external cause of this pathological condition. Other papers attribute bulk of lesions to overuse of horses by humans (riding). You instead find trauma a more likely cause it seems (warfare ?). Again, on what basis? What observed details suggest this?
Points 2, 3 and 4: First, thank you for your pertinent suggestions regarding the challenging lesion identified. All your suggestions were added in the body text of the manuscript, but some additional explanations are presented below. As the reviewer observed, we connected the identified lesion with an injury in the regional periosteum that triggered, eventually, periostitis. As known, in periostitis, only the outer layer of the bone, i.e. cortical bone, is affected as compared to osteomyelitis in which the bone marrow cavity or cancelous bone is involved too. In our case, only the outer part of the bone is affected as observed on x-ray and CT scan images. As a consequence of a rather chronic periosteal injury (chronic periostitis), the osteogenic layer of the periosteum reacts to insult forming ultimately the cancellous bone on top of the compact cortical bone as benign bony overgrowth, namely osteophytes (exostoses located on the joint margins) and exostoses at a distance from joints (as observed in our case). Since no muscle has significant insertion points in the hypertrophied zone (with osteophytes/exostoses), we presume the term enthesophyte is somehow unsuitable to use.
However, regarding the type of periosteal inflammation that could be responsible for benign bony overgrowth, suppurative periarthritis could be the one that triggered the periosteitis. Why do we suppose this?! Since the osteophytes and exostoses are situated all around the neck of the scapula (just dorsally to the glenoid cavity), the phlegmonous suppurative inflammation is the one with an invasive character (due to degrading enzymes of micro and macrophages), explaining this way the periostitis extending, probably, on both sides of the neck of the scapula. Since the glenoid cavity did not present any lesions, we believe the term arthritis should not be used, but suppurative periarthritis (causing later periostitis) is more suitable to the situation.
Concerning the etiology of the supurative periarthritis triggering eventually supraglenoidian periosteitis (basically the main cause of osteophytes/exostoses formation), the suppurative bacteria could arrive in the affected zone either by hematogenous way (from other sites than the leg region) or a regional onset may be another option. We prefer the last one because exactly over the affected region is passing the bursa of the infraspinate muscle. Is it a coincidence?! We suppose not…. Still, it is our deduction. Concerning the occurrence of suppurative bursitis of the infraspinate muscle, we believe the most likely way for bacterial seeding of the bursa could be by puncture or cut wound in the scapulohumeral joint region. Since no harness existed in that historical period overlapping the affected region, sharp utensils (e.g., arrowhead, spearhead) could be of preference.
- I do think, however, the paper is important and worth publishing. I just think more evidence needs to be given linking osteological observations to pathological internal and external causes.
Hope the previous 2 points responded to your concern. These ideas were inserted in the text
- I noted a few changes that should be made in attached document. Overall English fairly clear, though I sometimes got the impression standard English veterinary or anatomical terms were not properly used (see comments in paper).
Some notes were edited, and NAV names were inserted in many parts.
We saw your notes on the initial part- the anatomical description of structures and the overview of the common pathology and we would like to keep at least a shortened version of these parts, as we feel that, given the readership of this kind of materials (not mainly veterinarians, maybe biologists and historians), a brief review from anatomical/pathological perspective is a must and might orientate further investigation and literature research for them.
We have merged the two subheadings (anatomy and modern pathology) into a single one, leaving the paleopathology subheading distinct. If you still think that the removal of the anatomy part is necessary, we will do it for the next revision round.
All your notes were very helpful to us! Thank you very much for your careful and kind check-up!
Reviewer 4 Report
Comments and Suggestions for Authors
I think this manuscript is an interesting report centered on a specific finding. My main concern is the large amount of text reused by the authors from their recently-published paper "La Tène Horse Remains from Alba Iulia CX 143 Complex: A Whole Story to Tell", especially in the Introduction section. The authors should modify their Introduction section in order to make it original, adding specific information and background relevant to this manuscript and completely avoiding the use of parts of the text of other papers, even if those papers are theirs.
Comments on the Quality of English LanguageThere are some problems with the English and the authors should have a native speaker revise their manuscript.
Author Response
I think this manuscript is an interesting report centered on a specific finding. My main concern is the large amount of text reused by the authors from their recently-published paper "La Tène Horse Remains from Alba Iulia CX 143 Complex: A Whole Story to Tell", especially in the Introduction section. The authors should modify their Introduction section in order to make it original, adding specific information and background relevant to this manuscript and completely avoiding the use of parts of the text of other papers, even if those papers are theirs.
We made some changes to the introductory part, shortening and removing some of the mentioned data. We kept the main conclusions from our previous paper though, referencing the rest.
There are some problems with the English and the authors should have a native speaker revise their manuscript.
The text was checked once again by a professional. Several changes, both grammatical and semantical were operated. Some parts of the initial text were also removed and adapted to the reviewer’s demands
Thank you for your kindness and useful comments!
Reviewer 5 Report
Comments and Suggestions for Authors
The article “A paleopathological find on a La Tène horse skeleton discovered in rescue archaeological diggings in the area of the Olimpic Pool-Alba Iulia (CX 143 pit)” reports a particular pathological lesion found in a Late Iron Age (La Tène period) horse discovered in the CX 143 pit in Alba Iulia, Romania. Apparently, this is a first communication of such a case and therefore the paper represents a special interest. The authors provide a macroscopical and radiological study of the bone material and provide an interpretation of the malformation under study as periarticular and supraarticular hyperostosis. The article is well structured and written in good and clear language (however, I am not a native English speaker). The introduction provides necessary context of the study and of the archaeozoological material analyzed in the paper. The research method is clearly described. The description of material provides all necessary information about the histological nature of the malformation and provides high quality radiographic and CT images. The discussion and conclusions provide adequate interpretations of the obtained results. Generally, I have a very good impression about this paper and I didn’t find any flaws in the methodology, description, structure of the article, discussion and conclusions. I think this is the first time when I recommend to publish a paper as it is, without any objections. My congratulations to the authors.
Author Response
Thank you very much for your kind words. Appreciate! At the request of other reviewers, there were some changes made, also in respect to language and typos and also related to the content. Some improvements in the Results/Discussion part were made, as the Introductory part was slightly reduced.
Round 2
Reviewer 3 Report
Comments and Suggestions for Authors
Much improved, I just noted a few, most grammatical changes, that can be made [see attached document with marginal comments]. The discussion section now presents evidence to back up medical diagnosis in more detail.

Good job making improvements. See some comments in document as noted above.
Author Response
Dear Reviewer,
Thank you very much for your care and attention given to our paper. Your notes were fully considered and changes were made accordingly.
Once again, thanks a lot for your kindness!
Respectfully,
Alex Gudea